# A PINN Surrogate Modeling Methodology for Steady-State Integrated Thermofluid Systems Modeling

Kristina Laugksch [1,*] , Pieter Rousseau [1] and Ryno Laubscher [2]

1   Department of Mechanical Engineering, University of Cape Town, Private Bag X3, Rondebosch,
    Cape Town 7701, South Africa
2   Department of Mechanical and Mechatronic Engineering, Stellenbosch University, Private Bag X1, Matieland,
    Stellenbosch 7602, South Africa
*   Correspondence: lgkkri001@myuct.ac.za

**Abstract:** Physics-informed neural networks (PINNs) were developed to overcome the limitations associated with the acquisition of large training data sets that are commonly encountered when using purely data-driven machine learning methods. This paper proposes a PINN surrogate modeling methodology for steady-state integrated thermofluid systems modeling based on the mass, energy, and momentum balance equations, combined with the relevant component characteristics and fluid property relationships. The methodology is applied to two thermofluid systems that encapsulate the important phenomena typically encountered, namely: (i) a heat exchanger network with two different fluid streams and components linked in series and parallel; and (ii) a recuperated closed Brayton cycle with various turbomachines and heat exchangers. The results generated with the PINN models were compared to benchmark solutions generated via conventional, physics-based thermofluid process models. The largest average relative errors are 0.17% and 0.93% for the heat exchanger network and Brayton cycle, respectively. It was shown that the use of a hybrid Adam-TNC optimizer requires between 180 and 690 fewer iterations during the training process, thus providing a significant computational advantage over a pure Adam optimization approach. The resulting PINN models can make predictions 75 to 88 times faster than their respective conventional process models. This highlights the potential for PINN surrogate models as a valuable engineering tool in component and system design and optimization, as well as in real-time simulation for anomaly detection, diagnosis, and forecasting.

**Keywords:** deep learning; physics-informed neural networks; thermofluid process modeling

## 1. Introduction

Machine learning methods, such as neural networks, are typically employed in scientific fields to develop computationally efficient regression models with the capacity to fit high-dimensional, highly non-linear relationships accurately. Neural networks are emerging as valuable engineering tools for component and system design and optimization, as well as for real-time simulation for anomaly detection, diagnosis, and forecasting [1]. Conventional neural network models are developed using large datasets of training data. The data used to train such models are obtained from either experimental measurements or simulation results. However, the prohibitive cost and low fidelity of experimental data in industry-scale thermofluid systems, as well as the computational resources required in conventional simulation, limit the usefulness of purely data-driven machine learning methods for thermofluid process modeling [1,2]. Furthermore, purely data-driven neural network models are not useful for making predictions for scenarios that fall outside of the training range (i.e., extrapolating), therefore limiting their usefulness for thermofluid process modeling.

Physics-informed neural networks (PINNs) were developed to overcome the limitations of purely data-driven methods by embedding the physics equations directly into the

neural network loss function, thus circumventing the need for large databases of training data due to the physics-based regularization. The integration of prior knowledge, such as the governing physics equations, into the training process of the neural network model changes the training process from being supervised to being unsupervised. Thus far, PINNs have largely been applied to problems in the field of multi-dimensional computational fluid mechanics, but not to integrated thermofluid network modeling applications.

In the present work, the viability of applying a PINN surrogate modeling methodology to integrated thermofluid networks is explored using PINN models developed for two simple case studies. The scope of this work is limited to fully connected neural networks only. The purpose of this work is to investigate the performance of the PINN surrogate models compared to conventional, physics-based thermofluid process models. The solutions generated via the PINN models are therefore compared to benchmark solutions generated via conventional, physics-based thermofluid process models. A comparison will be made regarding the accuracy and computational speed of the two modeling methodologies.

Traditional thermofluid process models consist of physics-based methods that apply relevant first principle equations to develop numerical solvers. There are a variety of widely available industrial tools that apply such methods. For example, Ortega et al. [3] developed a thermofluid process model to evaluate the performance of a directly heated tubular solar receiver for an sCO2 Brayton cycle. The authors performed the required computational fluid dynamics (CFD) modeling using the Ansys Fluent® [4] fluid simulation software. This software applies numerical finite volume methods to solve the applicable partial differential equations (PDEs). Rauch et al. [5] employed a process model for a combined Brayton–Rankine cycle to determine the maximum thermal efficiency of the combined cycle. The process model consisted of a complete mathematical model and was developed using the Matlab® [6] programming platform, which is typically used for iterative analysis and design processes. The work of Zhang et al. [7] provides another example of a physics-based solver applied to thermofluid applications. The authors applied compartment models to analyze the reheat steam temperatures in a double reheat coal-fired boiler. This involved modeling the transfer of heat across different compartments, or control volumes within the boiler system. Although these traditional physics-based numerical solvers are accurate, they can be computationally expensive and time-consuming due to their complexity.

Recently, machine learning and deep learning techniques have been increasingly applied to develop computationally efficient models of complex processes. One commonly used technique is that of artificial neural networks (ANNs) in which many processing nodes, known as neurons, are organized into a network of interconnected layers. Conventionally, these ANNs are trained using large training datasets, and can thus be categorized as an example of a data-driven machine learning technique. Several authors have applied data-driven ANNs to a variety of thermofluid simulations. For example, Hosoz and Ertunc [8] developed an ANN to predict the performance of an automotive air conditioning (AAC) system. Haffejee and Laubscher [9] used a neural network to develop a data-driven surrogate model for an air-cooled condenser system at a power plant. Fast and Palmé [10] employed ANNs to develop a model for the online condition monitoring and diagnosis of a combined heat and power plant. In these examples, the trained ANNs were able to model the relevant systems with a high degree of accuracy while offering a significant increase in computational speed in comparison to conventional thermofluid process models.

Despite the benefits of using artificial neural networks over traditional thermofluid process models, several authors identified significant limitations of applying them to thermofluid problems. Pacheco-Vega et al. [11] developed feedforward neural networks to predict the performance of fin-tube heat exchangers used for refrigeration applications. They demonstrated that the predictive performance of the trained networks was directly linked to the size and distribution of the training data, with models built on undersized data performing poorly. This work, therefore, showed that the usefulness of data-driven surrogate modeling techniques is limited in applications where experimental or simulation data are not readily available, which is typical of industry-scale thermofluid systems.

Additionally, Willard et al. [12] note that ANNs can only provide reliable predictions for instances that are within the bounds of the training space. ANNs are thus not effective for scenarios where extrapolation beyond the training data is required.

PINNs were developed to overcome the challenges of traditional data-driven surrogate modeling approaches. Raissi et al. [2] first proposed the concept of PINNs in 2019 as a deep learning framework for solving complex physical systems. The authors used PINNs to construct accurate and computationally efficient surrogate models for complex partial differential equations across a variety of fields, such as the Schrödinger equation in quantum mechanics and the Navier–Stokes equations in fluid mechanics.

Since their introduction, PINNs have largely been applied to problems in the field of 1D, 2D, and 3D fluid mechanics. For example, Sun et al. [13] used PINNs to approximate the solutions to the Navier–Stokes equations to develop surrogate models of incompressible fluid flows for cardiovascular applications. The authors demonstrated that the application of a PINN methodology enabled the development of accurate surrogate models without the use of any labeled training data (i.e., CFD simulation data). Ang and Ng [14] applied PINNs to develop surrogate models for fluid flows around aerofoils at different angles of attack. The surrogate models were used to make predictions for the pressure and velocity fields. Once trained, the PINN surrogate models were able to generate results with comparable accuracy up to 4.5 times faster than conventional CFD solvers, thus demonstrating the significant reduction in computational cost that can be achieved with PINNs.

Zhu et al. [15] employed a physics-constrained learning approach to develop surrogate models for PDEs describing steady-state Darcy flow without labeled training data. The authors demonstrated that, given out-of-distribution test inputs, the generalization performance of the physics-constrained surrogate model was consistently better than that of the data-driven alternative. Cai et al. [16] reviewed the application of PINNs to problems in the field of fluid mechanics with reference to case studies that covered 3D incompressible flow, compressible flow, and biomedical flow. The authors demonstrated that PINNs are emerging as useful tools to simulate fluid flow for both forward problems, where the solutions to the governing equations are approximated, and inverse problems, where parameters characterizing the governing equation are extracted from the training data. Solving ill-posed inverse problems is beyond the reach of both traditional computational methods and purely data-driven machine learning methodologies.

Research into PINNs has gained significant momentum since their initial introduction, resulting in numerous recent advances in the methodology. Jagtap et al. [17] investigated the effect of using adaptive activation functions on the performance of PINNs used to model the Klein-Gordon, Helmholtz, and Burgers' equations. By introducing a scalable hyperparameter into the activation functions, the authors were able to adjust the gradient of the activation functions during training. In this way, the authors were able to increase the rate of convergence in comparison to the constant activation functions conventionally applied to neural networks. Wang et al. [18] applied the PINN methodology to a variety of applications, including the Helmholtz equation and flow in a lid-driven cavity. The authors proposed the use of a learning rate annealing algorithm to apply dynamic weights to the various terms constituting a given PINN loss function. These weights were used to balance the interplay between the different terms found in the neural network loss function. This approach improved both the trainability and predictive accuracy of the PINNs.

Despite the recent momentum of research into PINNs, they have yet to be applied to thermofluid process modeling applications (i.e., integrated 1D network modeling). No examples of PINNs applied to thermofluid process modeling applications have thus been observed in the literature.

In the present work, PINN models were developed for two integrated thermofluid systems, namely, a heat exchanger network and a simple recuperated supercritical carbon dioxide (sCO2) closed Brayton cycle. These models were created and trained on several singular samples using the *Python 3.10.6* and *TensorFlow 2.0* libraries to demonstrate the application of the proposed methodology.

## 2. Theoretical Background and Methodology

### 2.1. Multilayer Perceptron Neural Networks

Multilayer perceptron (MLP) networks are the most used category of artificial neural networks [19]. These networks can be trained to fit non-linear relationships between input and output variables accurately through exposure to known examples of corresponding data points. MLP networks consist of systems of interconnected signal processing nodes, known as neurons, that are divided into three sections, namely the input layer, hidden layers, and output layer, as shown in Figure 1. In a fully connected MLP network, each neuron in a given layer is connected to every other neuron in both the previous and subsequent layers. The number of neurons per hidden layer, as well as the number of hidden layers, are architecture hyperparameters that require tuning during the training process.

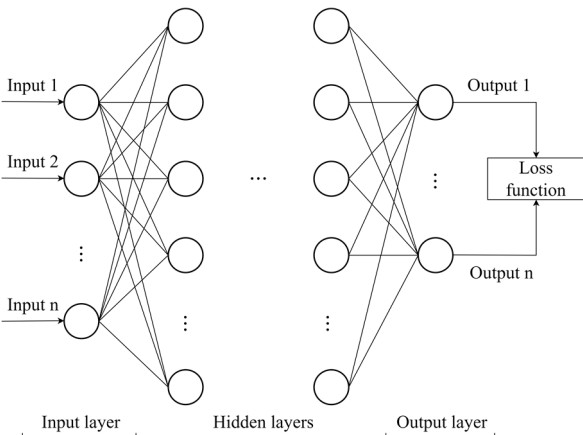

**Figure 1.** Typical architecture of a multilayer perceptron neural network.

To make a prediction, MLP networks feed the output from a neuron to each neuron in the downstream layer of the network. This output signal is multiplied by a weight and summed with the other incoming weighted signals. A bias value is added to this summed signal. To calculate the output signal of a given layer $\bar{a}_l$, the summed weighted incoming signal is then passed into a non-linear activation function. This process starts at the input layer and is repeated until the signal reaches the output layer. The output signal from the final layer is the predicted value of the MLP network. This process is known as forward propagation and is given in vector form as follows:

$$\bar{a}_l = \sigma_l \left( \bar{x}_{l-1} \cdot \bar{w}_l + \bar{b}_l \right) \tag{1}$$

In Equation (1), $\bar{x}_{l-1}$ is the output vector from the previous layer, $l-1$, $\bar{w}_l$ is a matrix containing all the connecting weights for the layer, $\bar{b}_l$ is a vector containing the layer biases, and $\sigma_l$ represents the activation function for the layer.

The network weights and biases constitute the trainable network parameters. These parameters are optimized to produce accurate predictions by minimizing a selected loss function. Typically, the mean squared error (MSE) between the predicted values $\hat{y}$ and the target values from the training dataset $Y$ is used as the loss function for MLP networks. The MSE loss function is given as:

$$MSE(\hat{y}, Y) = \frac{1}{n} \sum_{i=1}^{n} (\hat{y}_i - Y_i)^2 \tag{2}$$

The loss function is minimized using a gradient-based optimization process which requires the gradient of the loss function with respect to the trainable network parameters. With these gradients known, the trainable network parameters are updated iteratively to

minimize the loss function. The use of the given target values implies that MLP neural networks rely on supervised learning, as opposed to unsupervised learning.

### 2.2. PINN Methodology for Thermofluid Process Modeling

In the thermofluid network process modeling methodology, the system layout is described in terms of nodes and elements, as shown schematically in Figure 2.

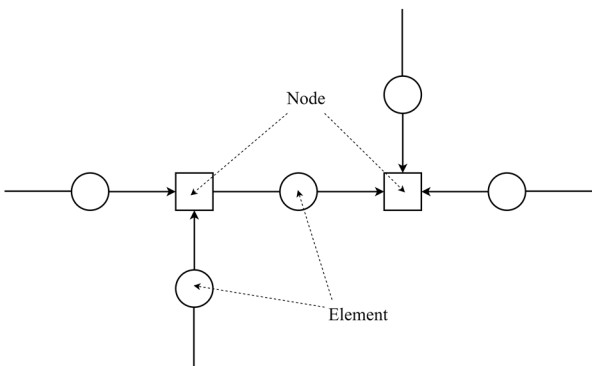

**Figure 2.** Thermofluid network represented by nodes and elements.

An element is a control volume that represents a physical component such as a pipe, valve, heat exchanger, boiler, or turbine. Each element has one inlet and one outlet and the fluid properties within the element are assumed to be represented by a single weighted average value between the inlet and the outlet. An element may also represent a single subdivision or increment of a physical component, such as a heat exchanger, that is discretized into several control volumes. A node represents the connection point between elements, which may also be a physical reservoir or tank. Nodes may therefore have multiple inlets and outlets with the fluid properties within a node assumed to be homogeneous and represented by a single averaged value.

MLPs form the foundation for PINNs, however, the loss functions for PINNs are constructed differently from those for MLPs. Instead of considering the difference between the predicted and target values, the governing physics equations for the system being modeled are embedded directly into the PINN loss function. For thermofluid process modeling applications, the governing physics equations are the mass, energy, and momentum balance equations. The mass flow rate ($\dot{m}[\text{kg}/\text{s}]$), total enthalpy ($h_0[\text{J}/\text{kg}]$), and total pressure ($p_0[\text{Pa}]$) are the fundamentally conserved quantities in the respective balance equations. These quantities are therefore the target parameters for which the PINN models generate predictions. The mass and energy balance equations are written for each node, while the momentum balance equation is written between the inlet and outlet of each element.

The generic forms of the balance equations are derived under the assumption of one-dimensional, steady-state flow through the network elements and are given by:

Mass balance (written for each node):

$$\Sigma\dot{m}_e = \Sigma\dot{m}_i \tag{3}$$

Energy balance (written for each node):

$$\Sigma\dot{m}_e h_{0e} = \Sigma\dot{m}_i h_{0i} + \dot{Q} - \dot{W} \tag{4}$$

Momentum balance (written for each element):

$$p_{0_e} = p_{0_i} + \Delta p_{0M} - \Delta p_{0L} \tag{5}$$

$\dot{Q}[\text{W}]$ is the rate of heat transfer to the fluid, $\dot{W}[\text{W}]$ is the rate of work done by the fluid, $\Delta p_{0M}[\text{Pa}]$ is the total pressure rise due to work done on the fluid, and $\Delta p_{0L}[\text{Pa}]$ is

the total pressure loss. For the purposes of the high-level integrated analysis that is the focus here, it will be assumed that the differences in the kinetic and potential energy terms between the inlets and outlets that form part of the total enthalpy and total pressure are negligible. This means that the total property values may be approximated with the static property values, and therefore $h_0 \approx h$ and $p_0 \approx p$ in Equations (4) and (5), respectively.

The values of the terms in the different balance equations are typically of different orders of magnitude and dependent on the specific application. For instance, one could find that $\dot{m} \approx 10^2$ kg/s, $\dot{m}h \approx 10^7$ W, and $p \approx 10^6$ Pa. These large differences will result in the energy balance equations providing larger contributions to the overall PINN loss function than the mass or momentum balance equations. As a result, the optimizer will be biased towards minimizing the disproportionately large energy balance losses during the training of the PINN, while neglecting the smaller mass and momentum losses [1]. This biased optimization process is highly undesirable as it ultimately leads to prolonged training times and inaccurate results. To prevent this, the conserved variables are normalized, and the balance equations are implemented in their non-dimensional (i.e., normalized) form. The normalized variables are defined as:

$$\dot{m}^* = \frac{\dot{m}}{\dot{m}_\infty}, p^* = \frac{p}{p_\infty}, h^* = \frac{h}{h_\infty} \tag{6}$$

In Equation (6), * denotes a non-dimensional variable. The variables are normalized using reference quantities (i.e., $\dot{m}_\infty, p_\infty, h_\infty$). The magnitudes of these reference quantities are selected such that they represented the maximum physically realistic values of the relevant variables for the specific thermofluid system.

The balance equations are applied throughout the thermofluid system to generate a set of residual functions. This entails writing the normalized balance equations with all quantities on one side of the equal sign. Using this approach, the generic residual loss functions for thermofluid systems are given as:

$$f_{mass} = \Sigma \dot{m}_e^* - \Sigma \dot{m}_i^* \tag{7}$$

$$f_{energy} = \Sigma \dot{m}_e^* h_e^* - \Sigma \dot{m}_i^* h_i^* - \frac{\dot{Q}}{\dot{m}_\infty h_\infty} + \frac{\dot{W}}{\dot{m}_\infty h_\infty} \tag{8}$$

$$f_{mom} = p_e^* - p_i^* - \frac{\Delta p_{0M}}{p\infty} + \frac{\Delta p_{0L}}{p\infty} \tag{9}$$

The different residual functions for the balance equations are then collected to form combined residual loss functions. The combined mass balance residual ($J_{mass}$), energy balance residual ($J_{energy}$), and momentum balance residual ($J_{mom}$) are calculated as mean squared losses and are given by:

$$J_{mass} = \frac{1}{N_{mass}} \sum_{i=1}^{N_{mass}} (f_{mass})^2 \tag{10}$$

$$J_{energy} = \frac{1}{N_{energy}} \sum_{i=1}^{N_{energy}} (f_{energy})^2 \tag{11}$$

$$J_{mom} = \frac{1}{N_{mom}} \sum_{i=1}^{N_{mom}} (f_{mom})^2 \tag{12}$$

The residual loss functions are then added together to obtain the overall PINN loss function as follows:

$$J_{loss} = \beta_1 J_{mass} + \beta_2 J_{energy} + \beta_3 J_{mom} \tag{13}$$

In Equation (13), $\beta_1$, $\beta_2$, and $\beta_3$ are user-defined weighting coefficients for the different residual loss functions.

The residual loss functions require the calculation of fluid properties and component characteristics for various thermofluid network components, such as specific work, specific heat, and pressure change, as inputs. The generic forms of the component characteristic equations are given as follows:

Total pressure rise due to work done on the fluid:

$$\Delta p_{0M} = p_{0i}(PR - 1) \tag{14}$$

with $PR = \frac{p_{0e}}{p_{0i}}$ the total pressure ratio over the machine.

Total pressure loss:

$$\Delta p_{0L} = \frac{K}{\rho}|\dot{m}|\dot{m} \tag{15}$$

with $\rho$ the average fluid density and $K$ a non-dimensional total pressure loss factor.

Rate of work done by the fluid:

$$\dot{W} = \eta\dot{m}(h_i - h_{es}) \tag{16}$$

with $\eta$ the isentropic efficiency of the machine and $h_{es}$ the isentropic outlet enthalpy.

Rate of heat transfer to the fluid:

$$\dot{Q}_h = \frac{\dot{Q}_{max,h}}{\left|\dot{Q}_{max,h}\right|}\varepsilon_{HX}\dot{Q}_{maxHX} \tag{17}$$

$$\dot{Q}_c = \frac{\dot{Q}_{max,c}}{\left|\dot{Q}_{max,c}\right|}\varepsilon_{HX}\dot{Q}_{maxHX} \tag{18}$$

with $\varepsilon_{HX}$ the heat exchanger's effectiveness, $\dot{Q}_{maxHX}$ the maximum possible rate of heat transfer, and the subscripts $h$ and $c$ referring to the hot and cold fluid streams, respectively.

In Equations (17) and (18), the maximum possible rate of heat transfer across the heat exchanger ($\dot{Q}_{maxHX}$) is given by:

$$\dot{Q}_{maxHX} = \min\left(\left|\dot{Q}_{max,h}\right|, \left|\dot{Q}_{max,c}\right|\right) \tag{19}$$

with the maximum possible rates of heat transfer for the hot and cold fluid streams, respectively, defined as:

$$\dot{Q}_{max,h} = \dot{m}\left(h_h|_{T_{ci}} - h_h|_{T_{hi}}\right) \tag{20}$$

$$\dot{Q}_{max,c} = \dot{m}\left(h_c|_{T_{hi}} - h_c|_{T_{ci}}\right) \tag{21}$$

In Equation (20), $h_h|_{T_{ci}}$ and $h_h|_{T_{hi}}$ are the enthalpy of the hot fluid at the inlet temperature of the cold fluid and the hot fluid, respectively, and in Equation (21) $h_c|_{T_{hi}}$ and $h_c|_{T_{ci}}$ are the enthalpy of the cold fluid at the inlet temperature of the hot fluid and the cold fluid, respectively.

In this work, the component characteristics are calculated within the PINN loss function. However, the required fluid properties are updated outside the loss function and then fed into the loss function for each evaluation. The fluid property calculations introduce additional non-linearities for which the PINN models need to account, over and above the set of non-linear balance and component characteristic equations. In this work, a bicubic interpolation scheme was applied to produce a set of non-linear interpolation functions capable of accurately approximating any required fluid properties given an input set of pressure and enthalpy values. The actual fluid properties used to generate these functions were obtained from the CoolProp fluid property library [20]. Figure 3 provides two ex-

amples of non-linear fluid property relationships that were captured using the bicubic interpolation scheme.

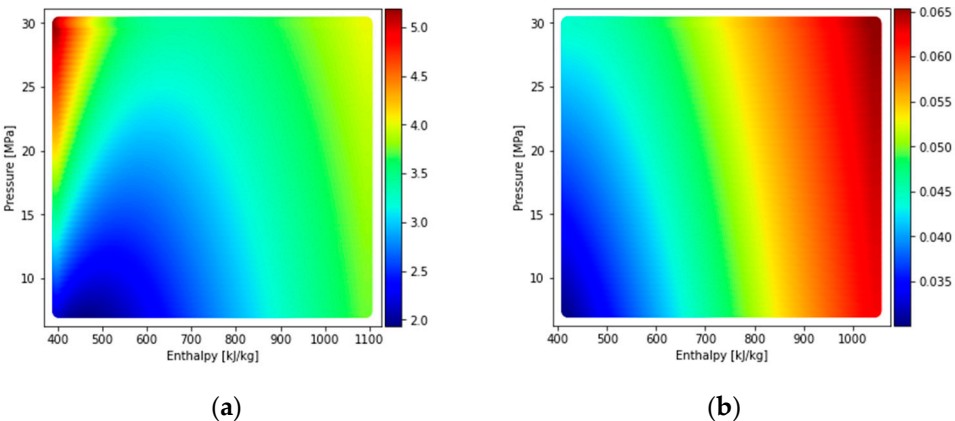

**Figure 3.** (**a**) Viscosity of carbon dioxide as a function of enthalpy and pressure; (**b**) Thermal conductivity of air as a function of enthalpy and pressure.

Besides the conserved variables ($\dot{m}, h, p$), there may also be other fluid properties (such as temperature $T[K]$) and/or component characteristic variables (such as a pressure loss coefficient $K$) that form part of the input features of the neural network. It is also important to normalize these parameters to obtain a normalized input feature vector in order to minimize the training time [21].

Two different optimization approaches were used to train the PINN model. One set of PINN models was trained using only the first-order Adam optimization algorithm [22] to update the network parameters. Adam was selected as the optimization algorithm for this work as it has been shown to outperform other optimization algorithms when training PINNs [23]. A second set of PINN models was trained using a hybrid optimizer in which the Adam optimizer was initially used to prime the parameters before a second-order optimizer was applied in conjunction with the Adam optimizer to train the network to convergence [24]. The truncated Newton method (TNC) was used as the second-order optimizer and implemented by coupling the *SciPy minimize* function with the neural network built in *TensorFlow*. When applying both a first- and second-order optimizer, the PINN model must first be trained using only the first-order optimizer for a set number of iterations before being refined using the second-order optimizer. This prevents the model from converging to a local minimum too quickly, as would be the case if the second-order optimizer was implemented on its own [24]. In this work, the first 400 iterations of the optimization process were completed using Adam before the TNC optimizer was applied.

In this work, the optimization process was primarily terminated based on a tolerance value for the total model loss. This ensured that the PINN models achieved a certain level of accuracy in their predictions. The value of $1 \times 10^{-6}$ was selected as the desired tolerance for the total model loss. A second restriction of a maximum number of iterations was placed on the optimization process of the PINN models to prevent the process from running indefinitely, should the models not be able to reach the desired tolerance specified above.

## 3. Case Studies

### 3.1. Heat Exchanger Network

In the present work, a simple heat exchanger network was modeled using the PINN methodology. Heat is transferred via various heat exchangers between two different fluid streams, namely air and supercritical carbon dioxide (sCO2). Despite its simplicity, the heat exchanger network encapsulates the important phenomena and complexities associated with thermofluid process modeling, as it requires the application of mass, energy, and momentum balances, two different fluids with heat transfer, and components configured in parallel and series. The layout of the heat exchanger network is shown in Figure 4.

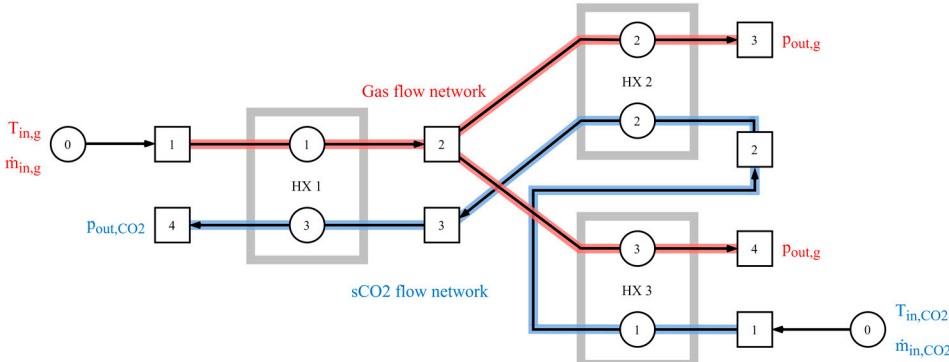

**Figure 4.** Thermofluid network model superimposed onto the physical layout of the heat exchanger system. Heat exchanger—HX.

The PINN model for the heat exchanger network predicts the enthalpies and pressures at each of the nodes, as well as the mass flow rates through each of the elements for both the gas and sCO2 flow networks. The input parameters for the network are given in Table 1. These parameters were varied to generate different sample points, thus simulating a variety of operating conditions.

**Table 1.** Input feature parameters for the heat exchanger network.

| Parameter | Symbol |
| --- | --- |
| Outlet pressures | $p_{out,g}, p_{out,CO2}$ |
| Inlet mass flow rates | $\dot{m}_{in,g}, \dot{m}_{in,CO2}$ |
| Inlet temperatures | $T_{in,g}, T_{in,CO2}$ |
| Heat exchanger lumped loss coefficients | $K_1, K_2, K_3$ |
| Heat exchanger effectiveness values | $\varepsilon_1, \varepsilon_2, \varepsilon_3$ |

The generic residual loss functions given in Equations (7)–(9) were applied to the heat exchanger network. For this case study, both the rate of work done by the fluid ($\dot{W}$) and the total pressure rise due to work done on the fluid ($\Delta p_{0M}$) are set to zero for all heat exchangers. Overall, 22 equations are solved simultaneously by minimizing the total combined residual loss.

The PINN model for the heat exchanger network uses the sigmoid activation function for the hidden layers and the linear activation function for the output layer. The definition of the sigmoid and linear activation functions are given in Equations (22) and (23), respectively.

$$\sigma_{Linear}(x) = \frac{1}{1 + e^{-x}} \tag{22}$$

$$\sigma_{Linear}(x) = x \tag{23}$$

The conventional process model for the heat exchanger network uses the set of governing equations for the system (i.e., the steady-state mass, momentum, and energy balance equations) written in a linearized form. The linearized equations were then used to construct two matrices, namely, one for the mass and momentum balance equations and another for the energy balance equations.

$$[X][Y] = [Z] \tag{24}$$

In Equation (24), $[X]$ represents a matrix of coefficients, $[Y]$ represents a vector of unknown quantities (e.g., unknown pressures), and $[Z]$ represents the source term vector. The set of linearized equations is then solved directly by inverting the coefficient matrices for the mass and momentum balance matrix as well as that for the energy balance matrix.

### 3.2. Recuperated Closed Brayton Cycle

In the present work, a simple recuperated supercritical carbon dioxide Brayton cycle was modeled using the PINN methodology. The simple sCO2 Brayton cycle was selected as an additional case study as it requires the consideration of different turbomachine components with associated non-linear performance characteristics, in addition to heat transfer processes. The process flow diagram for this cycle is shown in Figure 5.

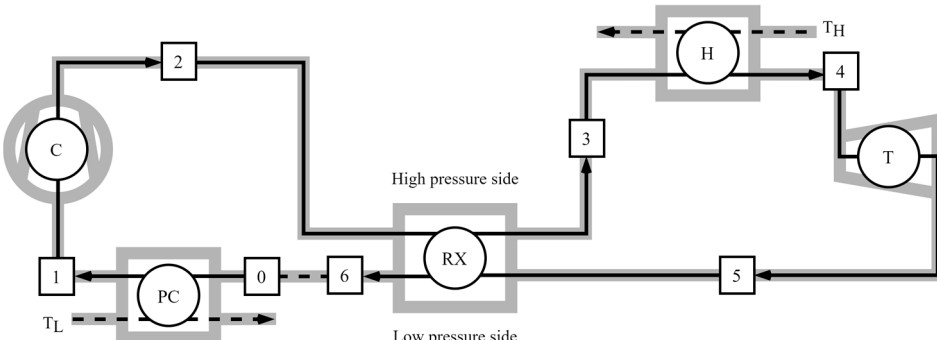

**Figure 5.** Process flow diagram for the recuperated Brayton cycle. Compressor—C, recuperator heat exchanger—RX, heater—H, turbine—T, pre-cooler—PC.

The PINN model for the recuperated Brayton cycle predicts the enthalpies and pressures at each of the nodes, as well as the total mass flow rates through the cycle. The input parameters for the recuperated Brayton cycle are given in Table 2. These parameters were varied to generate different sample points, thus simulating a variety of operating conditions.

**Table 2.** Input feature parameters for the recuperated Brayton cycle.

| Parameter | Symbol |
|---|---|
| Heat exchanger lumped loss coefficients | $K$ |
| Heat exchanger effectiveness values | $\varepsilon_{RX}, \varepsilon_H, \varepsilon_{PC}$ |
| Minimum cycle temperature | $T_L$ |
| Maximum cycle temperature | $T_H$ |
| Minimum cycle pressure | $p_L$ |

The generic residual loss functions given in Equations (7)–(9) were applied to the recuperated Brayton cycle. For this case study, both the rate of work done by the fluid ($\dot{W}$) and the total pressure rise due to work done on the fluid ($\Delta p_{0M}$) are set to zero for all heat exchangers, and the rate of heat transfer to the fluid ($\dot{Q}$) is set to zero for all turbomachinery. A set of 12 simultaneous equations is solved by minimizing the total combined residual loss.

The pressure ratio and efficiency versus mass flow rate characteristics of the turbo machines are represented via second-order polynomial curves. For the purposes of the present work, these curves were not obtained from actual performance data, but simply mimic the typical trends observed in real-world turbomachines. The coefficients for these polynomials are given in Table 3. The resulting component characteristic performance graphs for the compressor and turbine are shown in Figures 6 and 7, respectively. These polynomial curve fits introduce additional non-linearities which need to be accounted for by the PINN model. The PINN model for the recuperated sCO2 Brayton cycle uses the hyperbolic tangent activation function for the hidden layers and the linear activation

function for the output layer. The definition of the hyperbolic tangent and linear activation functions are given in Equations (25) and (23), respectively.

$$\sigma_{tanh}(x) = \frac{1 - e^{-2x}}{1 + e^{-2x}} \tag{25}$$

**Table 3.** Coefficients for the second-order polynomial machine performance curves.

| Compressor | | Turbine | |
|---|---|---|---|
| $a_{0C}$ | $3.947422$ | $a_{0T}$ | $-8.437242 \times 10^{-23}$ |
| $a_{1C}$ | $5.373162 \times 10^{2}$ | $a_{1T}$ | $3.313531 \times 10^{-9}$ |
| $a_{2C}$ | $-8.776627 \times 10^{5}$ | $a_{2T}$ | $9.730980 \times 10^{6}$ |
| $b_{0C}$ | $-1.942890 \times 10^{-16}$ | $b_{0T}$ | $1.054712 \times 10^{-15}$ |
| $b_{1C}$ | $1.453743 \times 10^{3}$ | $b_{1T}$ | $3.199740 \times 10^{3}$ |
| $b_{2C}$ | $-5.936429 \times 10^{5}$ | $b_{2T}$ | $-2.752242 \times 10^{6}$ |

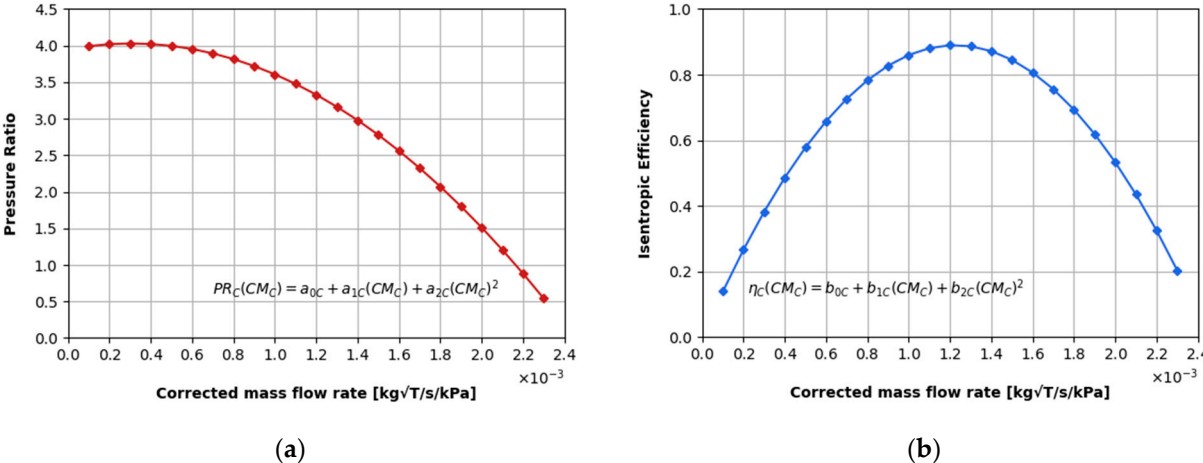

(**a**)  (**b**)

**Figure 6.** (**a**) Pressure ratio vs. corrected mass flow rate for the compressor; (**b**) Isentropic efficiency vs. corrected mass flow rate for the compressor.

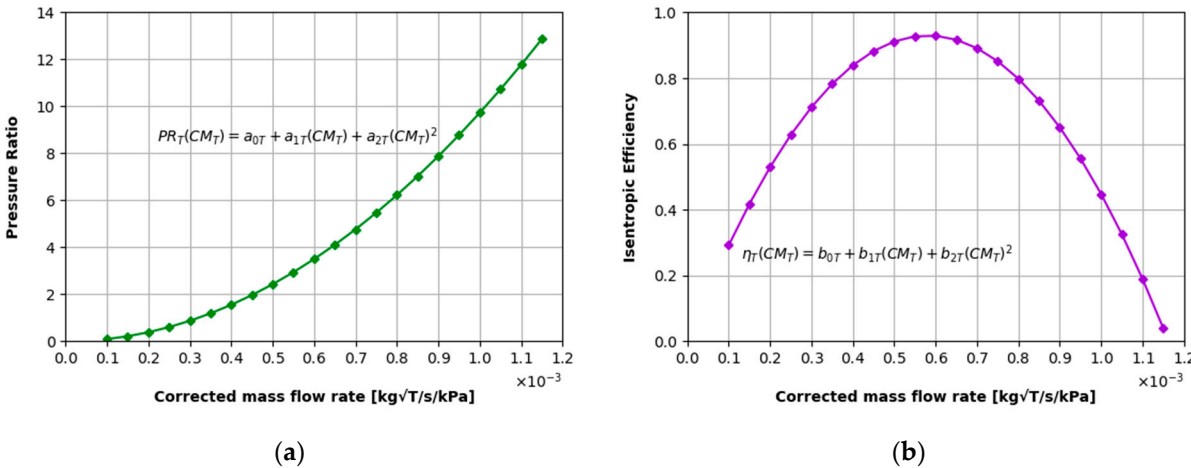

(**a**)  (**b**)

**Figure 7.** (**a**) Pressure ratio vs. corrected mass flow rate for the turbine; (**b**) Isentropic efficiency vs. corrected mass flow rate for the turbine.

The conventional process model for the heat exchanger network uses the set of governing equations for the system (i.e., the steady-state mass, momentum, and energy balance equations). This set of non-linear equations is solved simultaneously by iteratively applying a root-finding function until sufficient accuracy is obtained.

## 4. Results and Discussion

### 4.1. PINN Training Process

Initially, the weights for the two PINN models were randomly initialized using the Xavier procedure, while the network biases were all set to zero [25]. The heat exchanger PINN was trained from this randomly initialized state using only the balance equation loss function. Although this approach did result in convergence on some occasions, the majority of the predictions made during training were completely unrealistic, and the models failed to converge. For example, some predictions included negative values for both pressure and enthalpy. This can be attributed to the fact that the random initialization of network parameters resulted in the PINN model sometimes searching for a solution outside of the physically realistic domain.

To combat this, the trainable network parameters were preconditioned to predict physically realistic results by predicting prescribed values for all the mass flow rates, enthalpies, and pressures. This was achieved with a supervised pre-training step whereby the desired output values at all the nodes and elements throughout the thermofluid system were set equal to prescribed boundary values, and then training the PINN to predict these values. In effect, the PINN is pre-trained in a supervised mode to predict a trivial result in order to obtain realistic initial values for the trainable parameters. Using this two-step training approach, the PINN models consistently produced realistic results.

In the case of the heat exchanger network, the boundary values were the inlet mass flow rate and temperature, as well as the outlet pressures. In the case of the recuperated Brayton cycle, the boundary values were the minimum cycle temperature, as well as guessed values for the cycle mass flow rate and pressure.

### 4.2. Hyperparameter Search Results

A coarse grid search was implemented to find the best-performing fully connected PINN configurations for each of the two thermofluid networks. The hyperparameter search was implemented on the models which made use of the supervised pre-training step and used only the Adam optimizer during training, for which the learning rate was fixed at $1 \times 10^{-4}$ and $5 \times 10^{-5}$ for the heat exchanger network and the Brayton cycle, respectively.

The effects of the number of hidden layers and the number of neurons per hidden layer on the total model loss were investigated. For the heat exchanger network, the number of hidden layers was 1, 2, and 3. For each of these model depths, the number of neurons per hidden layer was 1, 8, 16, and 32. Similarly for the recuperated Brayton cycle, the number of hidden layers was 1, 2, and 3. However, for each of these model depths, the number of neurons per hidden layer was 1, 8, and 16. The results of the architecture hyperparameter search for the heat exchanger network and the recuperated Brayton cycle are shown in Tables 4 and 5, respectively.

The results in Tables 4 and 5 show that the total model loss can be reduced by increasing either the depth of the neural network or the number of neurons per hidden layer. However, the number of neurons per hidden layer had a more prominent effect on the total model loss than the depth of the neural network.

**Table 4.** Total model losses for different architectures for the heat exchanger network.

| Number of Neurons per Layer | Number of Hidden Layers | | |
|:---:|:---:|:---:|:---:|
| | **1** | **2** | **3** |
| 1 | $1.7 \times 10^{-1}$ | $1.2 \times 10^{-1}$ | $1.1 \times 10^{-1}$ |
| 8 | $2.6 \times 10^{-2}$ | $1.6 \times 10^{-2}$ | $1.2 \times 10^{-2}$ |
| 16 | $1.3 \times 10^{-3}$ | $1.0 \times 10^{-3}$ | $1.8 \times 10^{-4}$ |
| 32 | $6.5 \times 10^{-5}$ | $1.5 \times 10^{-5}$ | $5.7 \times 10^{-6}$ |

**Table 5.** Total model losses for different architectures for the recuperated Brayton cycle.

| Number of Neurons per Layer | Number of Hidden Layers | | |
|---|---|---|---|
| | 1 | 2 | 3 |
| 1 | $1.8 \times 10^{-1}$ | $1.1 \times 10^{-1}$ | $8.7 \times 10^{-2}$ |
| 8 | $3.2 \times 10^{-2}$ | $5.1 \times 10^{-3}$ | $2.3 \times 10^{-3}$ |
| 16 | $4.5 \times 10^{-5}$ | $4.7 \times 10^{-6}$ | $1.0 \times 10^{-6}$ |

In this work, the architecture configurations with the smallest total model loss values were selected. The final PINN model for the heat exchanger network, therefore, consists of three hidden layers with 32 neurons each, whereas the PINN for the recuperated Brayton cycle consists of three hidden layers with 16 neurons per layer.

*4.3. PINN Results: Accuracy*

In this work, two PINN models with different optimization approaches were developed for each case study as outlined in Section 2.2. A total of four PINN models were therefore developed. The PINN models for each of the case studies were trained and tested on 10 different samples which covered a range of operating conditions for the two thermofluid systems, such as different temperatures, pressures, and mass flow rates. The PINN models were trained and tested on each of these samples to demonstrate the range of validity of the PINN methodology. The statistical method known as Latin hypercube sampling was employed to ensure that the selected sample points sufficiently cover the full range of values possible within the design space [26]. The solutions generated by the trained PINN models for the different samples were compared against benchmark solutions that were generated using the conventional, physics-based thermofluid process models of the systems. The results of this comparison for the heat exchanger network are shown in Tables 6 and 7, and the results for the recuperated Brayton cycle are shown in Tables 8 and 9.

**Table 6.** Performance (absolute and relative errors) per output parameter for the PINN model of the heat exchanger network that used only Adam for optimization.

| | $\dot{m}_{CO2}$ [kg/s] | $h_{CO2}$ [kJ/kg] | $p_{CO2}$ [kPa] | $\dot{m}_{air}$ [kg/s] | $h_{air}$ [kJ/kg] | $p_{air}$ [kPa] |
|---|---|---|---|---|---|---|
| Maximum | 0.0052 | 2.3697 | 4.1911 | 0.0169 | 2.6047 | 0.1615 |
| Minimum | $2.98 \times 10^{-4}$ | $1.77 \times 10^{-1}$ | $3.51 \times 10^{-3}$ | $1.67 \times 10^{-3}$ | $9.78 \times 10^{-2}$ | $4.19 \times 10^{-3}$ |
| Average | 0.0030 | 1.4519 | 0.4699 | 0.0061 | 1.1724 | 0.0809 |
| Max (%) | 0.1227 | 0.2743 | 0.0182 | 0.3211 | 0.3124 | 0.1440 |
| Min (%) | $5.70 \times 10^{-3}$ | $2.21 \times 10^{-2}$ | $2.39 \times 10^{-5}$ | $3.03 \times 10^{-2}$ | $1.25 \times 10^{-2}$ | $3.84 \times 10^{-3}$ |
| Avg (%) | 0.0611 | 0.1745 | 0.0020 | 0.1165 | 0.1421 | 0.0738 |

**Table 7.** Performance (absolute and relative errors) per output parameter for the PINN model of the heat exchanger network that used the hybrid Adam-TNC optimizer.

| | $\dot{m}_{CO2}$ [kg/s] | $h_{CO2}$ [kJ/kg] | $p_{CO2}$ [kPa] | $\dot{m}_{air}$ [kg/s] | $h_{air}$ [kJ/kg] | $p_{air}$ [kPa] |
|---|---|---|---|---|---|---|
| Maximum | 0.0057 | 1.9053 | 2.8155 | 0.0117 | 1.6225 | 0.1079 |
| Minimum | $8.97 \times 10^{-5}$ | $6.34 \times 10^{-1}$ | $7.16 \times 10^{-3}$ | $5.19 \times 10^{-4}$ | $7.24 \times 10^{-1}$ | $1.19 \times 10^{-2}$ |
| Average | 0.0031 | 1.3730 | 0.3685 | 0.0055 | 1.1017 | 0.0681 |
| Max (%) | 0.1442 | 0.2683 | 0.0133 | 0.2700 | 0.1946 | 0.0966 |
| Min (%) | $1.71 \times 10^{-3}$ | $7.95 \times 10^{-2}$ | $4.88 \times 10^{-5}$ | $9.02 \times 10^{-3}$ | $9.22 \times 10^{-2}$ | $1.15 \times 10^{-2}$ |
| Avg (%) | 0.0648 | 0.1662 | 0.0018 | 0.1109 | 0.1351 | 0.0618 |

**Table 8.** Performance (absolute and relative errors) per output parameter for the PINN model of the recuperated Brayton cycle that used only Adam for optimization.

|           | $\dot{m}$ [kg/s] | $h$ [kJ/kg] | $p$ [kPa] |
|-----------|--------|--------|---------|
| Maximum   | 7.046  | 22.389 | 171.208 |
| Minimum   | 0.420  | 0.417  | 19.630  |
| Average   | 3.162  | 6.379  | 85.745  |
| Max (%)   | 1.127  | 3.128  | 0.803   |
| Min (%)   | 0.0813 | 0.0609 | 0.1062  |
| Avg (%)   | 0.5353 | 0.9266 | 0.4545  |

**Table 9.** Performance (absolute and relative errors) per output parameter for the PINN model of the recuperated Brayton cycle that used the hybrid Adam-TNC optimizer.

|           | $\dot{m}$ [kg/s] | $h$ [kJ/kg] | $p$ [kPa] |
|-----------|--------|--------|---------|
| Maximum   | 6.753  | 22.133 | 145.228 |
| Minimum   | 1.013  | 0.448  | 19.887  |
| Average   | 2.920  | 6.155  | 80.600  |
| Max (%)   | 1.080  | 3.092  | 0.767   |
| Min (%)   | 0.171  | 0.065  | 0.108   |
| Avg (%)   | 0.499  | 0.893  | 0.429   |

The results shown in Tables 6 and 7 illustrate that both the PINN models for the heat exchanger network were able to produce results for the mass flow rates, pressures, and enthalpies that are within 0.17% of the benchmark solutions. There is no significant difference in the accuracy of the two optimization approaches. Similarly, the results shown in Tables 8 and 9 illustrate that both the PINN models were able to make predictions for the recuperated Brayton cycle that are in close agreement with the benchmark solutions. The two PINN models produced average relative errors that are less than 0.93% for the mass flow rates, pressures, and enthalpies. Once again, there is no significant difference in accuracy between the two optimization approaches.

These results illustrate that the PINN modeling methodology can be applied to model thermofluid systems to generate accurate predictions. Interestingly, the PINN models of the heat exchanger network produced predictions with lower relative errors than the models of the recuperated Brayton cycle. This can likely be attributed to the additional non-linearities present in the Brayton cycle as a result of the inclusion of the turbomachine performance characteristics.

*4.4. PINN Results: Computational Expense*

Figures 8 and 9 provide training histories for the unsupervised training processes of the four different PINN models. The training histories show that, on average, the PINN models that used the hybrid Adam-TNC optimizer converged to an acceptable tolerance in fewer iterations than the models that used only the Adam optimizer. This is true for both the heat exchanger network and the recuperated Brayton cycle. This finding is corroborated by the results shown in Table 10 which provide a summary of the number of training iterations each PINN model required to reach convergence. The use of the second-order optimizer reduced the average number of iterations by approximately 180 in the case of the heat exchanger and 690 in the case of the Brayton cycle. This demonstrates that, in general, the use of the hybrid Adam-TNC optimization approach reduces the computational expense of training PINNs.

Furthermore, the PINN model for the Brayton cycle that used the hybrid Adam-TNC optimizer was able to reach the required tolerance for all samples within the maximum number of iterations, whereas the model that only used the Adam optimizer was unable to do so for one of the samples. The hybrid Adam-TNC optimization approach is therefore the more desirable approach for training PINN models for the two integrated thermofluid systems.

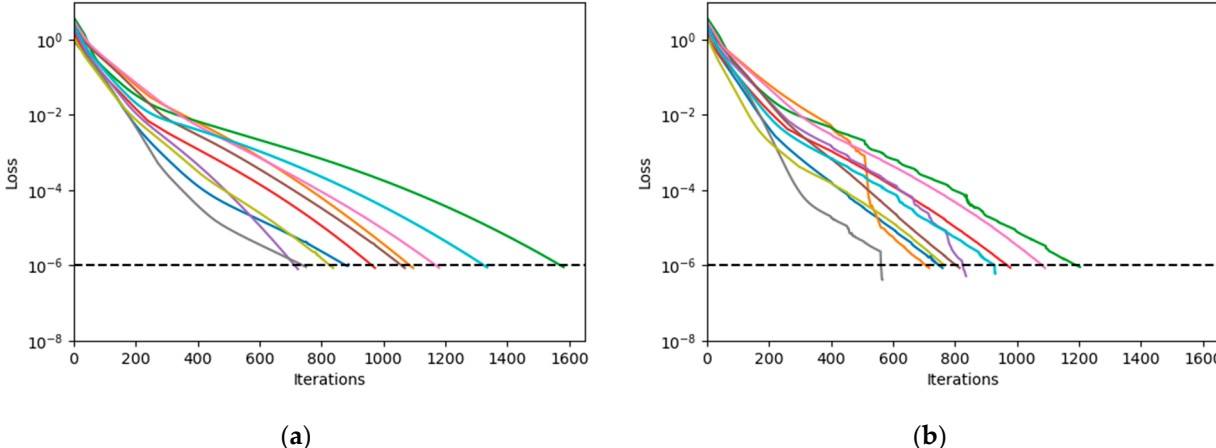

(a)                                                         (b)

**Figure 8.** (**a**) Training history for the unsupervised training of the PINN model of the heat exchanger network using only the Adam optimizer; (**b**) Training history for the unsupervised training of the PINN model of the heat exchanger network using a combination of the Adam optimizer and the truncated Newton method. The various colors each represent a different sample of the same simulation.

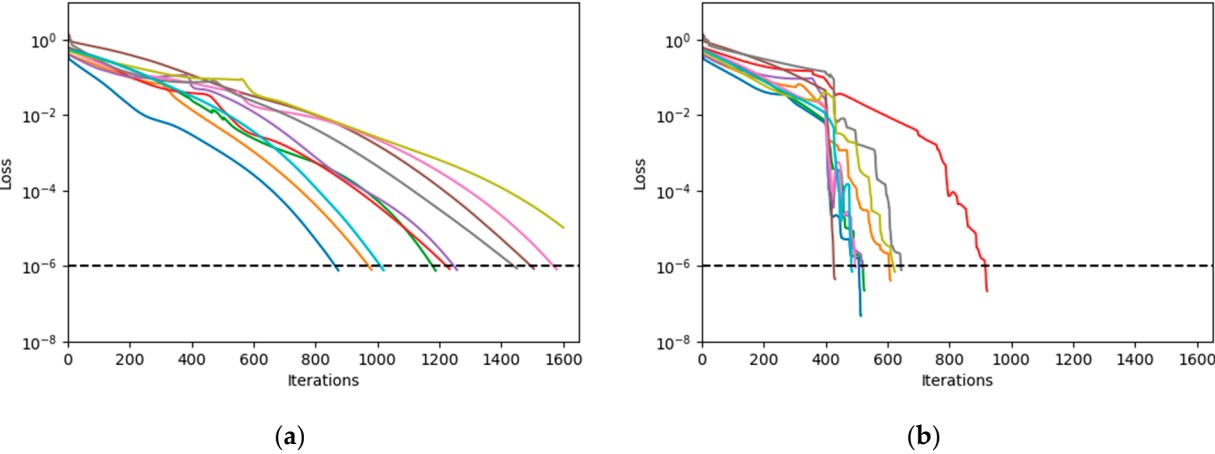

(a)                                                         (b)

**Figure 9.** (**a**) Training history for the unsupervised training of the PINN model of the recuperated Brayton cycle using only the Adam optimizer; (**b**) Training history for the unsupervised training of the PINN model of the recuperated Brayton cycle using a combination of the Adam optimizer and the truncated Newton method. The various colors each represent a different sample of the same simulation.

**Table 10.** Number of iterations for the various PINN models.

|  | Heat Exchanger Network | | Brayton Cycle | |
|  | Adam Only | Adam and TNC | Adam Only | Adam and TNC |
|---|---|---|---|---|
| Max | 1582 | 1203 | 1600 | 922 |
| Min | 724 | 566 | 874 | 431 |
| Avg | 1044 | 866.1 | 1268.7 | 577.6 |

Once trained, the PINN models were able to make predictions for the two thermofluid systems in a fraction of the time taken by the conventional process models, as is shown in Table 11. On average, the PINN model of the heat exchanger made predictions 75 times faster than the conventional process model for the same system, whereas the PINN model of the Brayton cycle made predictions 88 times faster than the conventional process model for the same system. These results demonstrate that the PINN modeling methodology enables significant reductions in computational time in comparison to conventional process

modeling methods. It is therefore likely that PINNs will offer significant reductions in computational expense when the problem is scaled up to include large data samples for surrogate modeling. Furthermore, it is likely that PINNs will successfully be applied to thermofluid problems that require real-time simulation.

**Table 11.** Time taken to generate a solution.

| | Heat Exchanger Network | | Brayton Cycle | |
| | Trained PINN | Conventional Process Model | Trained PINN | Conventional Process Model |
| --- | --- | --- | --- | --- |
| Max (s) | 0.0156 | 0.5369 | 0.0092 | 0.5690 |
| Min (s) | 0.0018 | 0.2890 | 0.0010 | 0.1396 |
| Avg (s) | 0.0053 | 0.3976 | 0.0033 | 0.2904 |

## 5. Conclusions

It was shown that a PINN modeling methodology can be applied successfully to model integrated thermofluid systems using the non-dimensionalized forms of the mass, energy, and momentum balance equations in the loss function. The trainable network parameters must first be trained using a supervised pre-training step before the PINN loss function is implemented to ensure consistent performance and accurate predictions from the PINN models. Furthermore, it was shown that the use of a hybrid Adam-TNC optimizer provided a significant computational advantage over a pure Adam optimization approach, as PINN models trained with this approach required significantly fewer training iterations to reach convergence.

It was shown that the time taken for the trained PINN models to make a prediction is in the order of $1 \times 10^{-3}$ s. This is a significant improvement over the conventional process models which required $1 \times 10^{-1}$ s to generate a solution. If these modeling approaches are to be applied to a more complex thermofluid network, the time taken for the conventional model to generate a solution will scale non-linearly and it is likely that it would require approximately $1 \times 10^1$ s to generate a solution. By contrast, the inference speed of PINNs remains constant, regardless of the complexity of the network, provided that the GPU RAM is not exceeded [27]. This means that for a problem that requires 10 model calls it would take $1 \times 10^2$ s to generate a solution using a conventional process model, but only $1 \times 10^{-2}$ s using a PINN model. This highlights the potential for PINN surrogate models as a valuable engineering tool in component and system design and optimization, as well as in real-time simulation for anomaly detection, diagnosis, and forecasting.

Further work will entail the expansion of the PINN modeling methodology to multiple simultaneous samples to investigate the capacity of PINNs to be used for interpolation and extrapolation as well as surrogate modeling. Furthermore, future work could also investigate the effect of different network structures, such as graph convolutional neural networks and convolutional structures, on the PINN loss function.

**Author Contributions:** Conceptualization, K.L., P.R. and R.L.; methodology, K.L., P.R. and R.L.; software, K.L.; formal analysis, K.L.; writing—original draft preparation, K.L.; writing—review and editing, P.R. and R.L.; visualization, K.L.; supervision, P.R. and R.L.; funding acquisition, K.L. and P.R. All authors have read and agreed to the published version of the manuscript.

**Funding:** This work is based on research supported by the National Research Foundation of South Africa (Grant Numbers 138618 and 148757) and the University of Cape Town.

**Data Availability Statement:** The data presented in this study are available on request from the corresponding author.

**Conflicts of Interest:** The authors declare no conflict of interest.

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
