# Peer review of "A PINN Surrogate Modeling Methodology for Steady-State Integrated Thermofluid Systems Modeling"

_mca, doi:10.3390/mca28020052_

Round 1

Reviewer 1 Report

Report for: A PINN Surrogate Modelling Methodology for Steady-State Integrated Thermofluid Systems Modelling

This paper proposes a PINN surrogate modelling methodology for steady-state integrated thermofluid systems modelling based on the mass, energy, and momentum balance equations, combined with the relevant component characteristics and fluid property relationships.

In summary, I feel that this manuscript is good and well-organized work, and is suitable for publication. Thus, I am glad to recommend its acceptance for publication in Mathematical and Computational Applications if the following suggestions are addressed.

1. The manuscript shows the total model losses for neural networks with different structures, l wonder whether the neural network with deeper hidden layers or more neurons in per hidden layer will affect the loss function.

2. The author claimed that “The PINN models were trained and tested on 10 different samples”, What are the characteristics and differences of each sample? The purpose of using different sample to train neural networks is not shown in the manuscript.

3. Whether the Adam optimizer can be replaced by other optimizers?

Reviewer 2 Report

The manuscript is interesting, and the topic is worth investigating. However, I have some comments that should be addressed before publication.

The authors discuss fluid dynamics problems, the applied case study-related literature, and neural network-related development. However, I miss the mention of compartment models, which I think also relates to this field and has similar computation time 'sparing' than the proposed method (or better) and is also closely related to the physical system. Please discuss compartment models shortly.

Figure 2 is not clean; please replace it with a sharper image.

Please discuss the validation results in more detail, and describe the model benchmark used for validation.

The results are clearly interesting, and the method is working. However, both case studies can be solved using algebraic equations (obviously not as precise as CFD), precise enough for engineering calculations, or flowsheeting simulators can be used for this task with relatively minimal computation time. The authors should highlight the superiority of their method and justify its use of it.
